# Multi-Objective Swarm Intelligence Trajectory Generation for a 7 Degree of Freedom Robotic Manipulator

**Aryslan Malik \*** **, Troy Henderson and Richard Prazenica**

Aerospace Engineering Department, Embry—Riddle Aeronautical University, Daytona Beach, FL 32114, USA;
hendert5@erau.edu (T.H.); prazenir@erau.edu (R.P.)
* Correspondence: malika3@my.erau.edu

**Abstract:** This work is aimed to demonstrate a multi-objective joint trajectory generation algorithm for a 7 degree of freedom (DoF) robotic manipulator using swarm intelligence (SI)—product of exponentials (PoE) combination. Given a priori knowledge of the end-effector Cartesian trajectory and obstacles in the workspace, the inverse kinematics problem is tackled by SI-PoE subject to multiple constraints. The algorithm is designed to satisfy finite jerk constraint on end-effector, avoid obstacles, and minimize control effort while tracking the Cartesian trajectory. The SI-PoE algorithm is compared with conventional inverse kinematics algorithms and standard particle swarm optimization (PSO). The joint trajectories produced by SI-PoE are experimentally tested on Sawyer 7 DoF robotic arm, and the resulting torque trajectories are compared.

**Keywords:** PoE; machine learning; swarm; robot-manipulation; inverse kinematics; trajectory generation



## 1. Introduction

Trajectory generation and motion planning is an important part of robot control, which most often is carried out with end-effector's position and orientation in mind. This is not problematic when the closed-form analytical solution is available. However, in cases where there is no such solution, the process of obtaining joint trajectories or inverse kinematics (IK) becomes a challenging task, especially in the presence of obstacles or when the effort minimization is of importance as well. The inverse kinematics (IK) problem has been a hot topic in robotics field for a long time, and many different approaches were demonstrated to generate joint trajectories that satisfy a specific end-effector Cartesian trajectory. As the agility of robotic manipulators becomes a crucial design consideration, which increases the number of joints, the IK problem becomes even more involved as redundancy is introduced. Thus, opting for a machine learning (ML), artificial neural networks (ANNs), or SI algorithms to handle such highly non-linear problem looks very attractive, which is evident by the recent interest in using SI/PSO algorithms to tackle the IK problem.

Generally, the algorithms solving the IK problem can be classified into two categories: pseudo-inverse Jacobian and iterative non-linear root finding methods, and ANN/ML/SI methods. The first category can be considered as the conventional method of approaching the IK problem, where the algorithm most often aims to solve for the joint trajectory while satisfying only a single objective, such as minimizing joint effort and/or movement. Some works proposed using Jacobian pseudo-inverse methods to prioritize tasks in workspace [1] and keep the joint limits within the physical bounds [2]. Amongst iterative approaches, the most often appearing is the Newton–Raphson iterative non-linear root-finding method which considers the IK problem as a non-linear optimization problem [3]. One recent study explored the idea of searching suboptimal paths using graph theory and Dijkstra algorithm [4] which minimized the movement time between the given positions while avoiding collisions with the obstacles at the same time [5].

The latter category can be described as a fresh view on the IK problem, since, in this approach, techniques that were only recently developed and applied to engineering problems are leveraged. Various metaheuristic methods, such as genetic algorithms (GA) and SI have proven to be effective in solving IK even for robotic manipulators with high DoFs. For instance, procedural, non-linear, and multi-modal GA with parallel populations and migration technique was implemented to produce time-optimal trajectory planning for hyper-redundant manipulators [6]. Deep deterministic policy gradient (DDPG) and normalized advantage function (NAF) algorithms have shown their usefulness in continuous action spaces and, more specifically, in robotic manipulation [7,8]. Deep Q-Networks (DQN) proved their usefulness in the robotics field, where its architecture was used for vision-based manipulation [9,10], path planning [11,12], navigation [13,14], IK solution for a high-DoF robotic systems [15–17], and even collision avoidance [18]. This work focuses on developing a SI algorithm satisfying multiple objectives at once. Amongst SI techniques, the particle swarm optimization (PSO) received the most attention due to its performance on high-DoF IK problems [19]. Plenty of the literature is available on different variations of PSO applied to IK problems. One work's approach was to decouple the manipulator into two segments, thus approaching the IK in a bidirectional fashion [20]. Another work decoupled position and orientation by applying two PSO algorithms to achieve faster convergence, and, in addition to that, used inverse Jacobian to smooth the trajectories, thus achieving position control [21]. Attempts to improve artificial algorithm's performance were made by adding constriction factor and adaptive inertia to the PSO [22], applying non-linear dynamic inertia weight adjustment [23], and even combining PSO with Agoraphilic for obstacle avoidance [24]. The scalability of the PSO to high-DoF IK problems was explored [19]. Furthermore, a quantum-behaved PSO was proposed as an IK solution where it showed an improvement in performance [25].

Even though many works have been published with different SI/PSO variants, there are certain aspects that most often were left obscured such as unclear indication of how exactly inverse kinematics problem was set up and solved, unclear collision identification algorithm, absence of torque trajectories, absence of initial conditions for each SI/PSO iteration, absence of error in end-effector position/orientation, application of the algorithm on planar (2-D) manipulators, experimental evaluation almost exclusively in simulation, etc. The objective of this work is to address the aforementioned shortcomings by demonstrating a novel SI-PoE algorithm and experimentally validating it by applying it directly to 7 DoF Sawyer robotic manipulator. It should be noted that this algorithm is not limited to the Sawyer robotic arm and can be generalized to any n-DoF robotic system, either by defining screw axes and home configuration or, if Denavit–Hartenberg (D-H) parameters are known, the kinematics model can be transformed to the PoE formulation as outlined in [3]. The SI-PoE algorithm is used for the cases when the end-effector trajectory and obstacles are defined a priori in the workspace. The main idea is that the IK problem is solved by the SI, given multiple goals such as minimizing control effort, avoiding obstacles, and enforcing finite jerk on the end-effector. An additional quintic polynomial finite jerk (QPFJ) method of trajectory generation is also explored to demonstrate the possiblity to enforce finite jerk on joint space. The PoE is used as means of identifying any collisions with the obstacles in the workspace. However, it should be noted that most often these goals overconstrain the solution leading to cases where a trade-off has to be made; for example, the end-effector Cartesian trajectory accuracy might suffer for some cases if finite jerk hard constraint is imposed on the joints.

The rest of the paper is outlined as follows: Section 2 describes the robotic arm, finite jerk end-effector trajectory generation, and the PoE forward kinematics (PoE-FK) algorithm used in setting up an IK problem and fitness function for SI. Section 3 outlines the SI-PoE algorithm and its fitness function along with parameters, collision detection mechanism, particles' initial conditions, and computational performance. Section 4 presents the resulting joint trajectories, methods used to smooth them, and error in end-effector's position. Section 5 provides the simulated and experimental torque profiles and discussion

on how the SI-PoE parameteres can be varied case by case for optimal performance. Finally, Section 6 concludes the work with final remarks.

## 2. Robotic Arm Description, End-Effector Trajectory, and PoE-FK

The robotic arm used in the simulation and experiments is Rethink Robotics' 7-revolute (7R) Sawyer robot, which is shown in Figure 1, where the robotic arm is at its home configuration with all joint positions at zero ($\theta_1 = 0, \theta_2 = 0, \cdots, \theta_7 = 0$). Home configuration and all of the dimensions were taken from universal robot description format (URDF) file dedicated to the Sawyer robot [26], and simple geometry reconstructed in Matlab is shown in Figure 2. Sawyer has 7 links and 7 revolute joints, 4 of which are rolling and 3 are pitching joints.

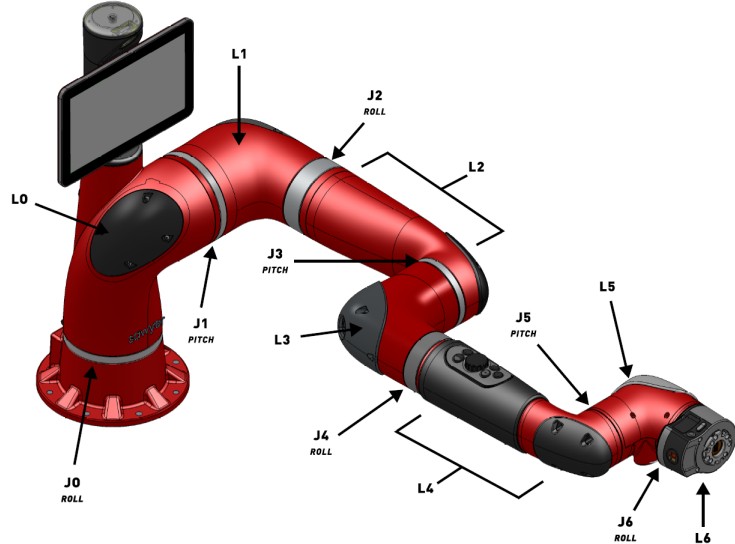

**Figure 1.** Sawyer joints and links.

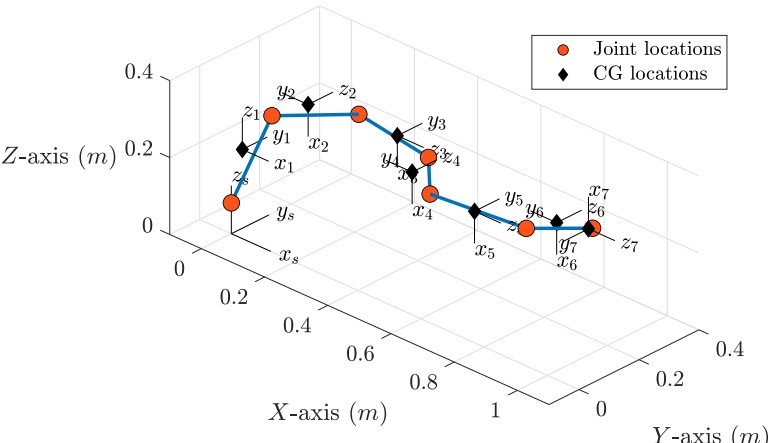

**Figure 2.** Home configuration of the Sawyer robotic arm represented in Matlab.

Forward kinematics is realized using PoE-FK. This method was chosen because it enjoys certain advantages over a conventional D-H parameters approach, which includes but is not limited to intuitive geometric interpretation that leads to easier set up process, uniform treatment of revolute and prismatic joints, absence of strict rules to assign frames, and concise and elegant formula [27,28].

Given physical locations of joints in home configuration from URDF file, the screw axes in the space frame can be shown in Equations (1) and (2).

$$
\mathcal{S}_1 = \begin{bmatrix} 0 \\ 0 \\ 1 \\ 0 \\ 0 \\ 0 \end{bmatrix} \qquad
\mathcal{S}_2 = \begin{bmatrix} 0 \\ 1 \\ 0 \\ -0.317 \\ 0 \\ 0.081 \end{bmatrix} \qquad
\mathcal{S}_3 = \begin{bmatrix} 1 \\ 0 \\ 0 \\ 0 \\ 0.317 \\ -0.1925 \end{bmatrix} \qquad
\mathcal{S}_4 = \begin{bmatrix} 0 \\ 1 \\ 0 \\ -0.317 \\ 0 \\ 0.481 \end{bmatrix} \tag{1}
$$

$$
\mathcal{S}_5 = \begin{bmatrix} 1 \\ 0 \\ 0 \\ 0 \\ 0.317 \\ -0.024 \end{bmatrix} \qquad
\mathcal{S}_6 = \begin{bmatrix} 0 \\ 1 \\ 0 \\ -0.317 \\ 0 \\ 0.881 \end{bmatrix} \qquad
\mathcal{S}_7 = \begin{bmatrix} 1 \\ 0 \\ 0 \\ 0 \\ 0.317 \\ -0.1603 \end{bmatrix} \tag{2}
$$

The PoE-FK formula, which represents the position and orientation of a frame (point) attached to $n$-th link, is shown below:

$$
T_{sn} = e^{[\mathcal{S}_1]\theta_1} e^{[\mathcal{S}_2]\theta_2} \cdots e^{[\mathcal{S}_n]\theta_n} M_{sn} \tag{3}
$$

where, $M_{sn} \in SE(3)$ is a frame (position and orientation) attached to the robotic arm's $n$-th link given in home configuration. Since the screw axis is a normalized twist, the skew-symmetric representation $[\mathcal{S}]$ of $\mathcal{S} = (\omega, v)$ is:

$$
[\mathcal{S}] = \begin{bmatrix} [\omega] & v \\ 0 & 0 \end{bmatrix} \in se(3), \quad
[\omega] = \begin{bmatrix} 0 & -\omega_3 & \omega_2 \\ \omega_3 & 0 & -\omega_1 \\ -\omega_2 & \omega_1 & 0 \end{bmatrix} \in so(3) \tag{4}
$$

Thus, the matrix exponential mapping becomes:

$$
e^{[\mathcal{S}]\theta} = \begin{bmatrix} e^{[\omega]\theta} & (I\theta + (1 - \cos\theta) + (\theta - \sin\theta)[\omega]^2)v \\ 0 & 1 \end{bmatrix} \in SE(3) \tag{5}
$$

where exponential $e^{[\omega]\theta}$ comes from Rodrigues' formula for rotations:

$$
e^{[\omega]\theta} = I + \sin\theta[\omega] + (1 - \cos\theta)[\omega]^2 \in SO(3) \tag{6}
$$

The complete derivation of the PoE-FK can be found in our previous work [29]. Now, the choice of frames (points) to be tracked by PoE-FK is of paramount importance and is up to the user. For example, if the center of gravity (CG) of each link has to be tracked, the following matrices extracted from URDF file can be used:

$$
M_{s1} = \begin{bmatrix} 1 & 0 & 0 & 0.0244 \\ 0 & 1 & 0 & 0.0110 \\ 0 & 0 & 1 & 0.2236 \\ 0 & 0 & 0 & 1 \end{bmatrix} \qquad
M_{s2} = \begin{bmatrix} 0 & -1 & 0 & 0.1078 \\ 0 & 0 & 1 & 0.1425 \\ -1 & 0 & 0 & 0.3201 \\ 0 & 0 & 0 & 1 \end{bmatrix} \tag{7}
$$

$$
M_{s3} = \begin{bmatrix} 0 & 0 & 1 & 0.3568 \\ 0 & 1 & 0 & 0.1775 \\ -1 & 0 & 0 & 0.3172 \\ 0 & 0 & 0 & 1 \end{bmatrix} \qquad
M_{s4} = \begin{bmatrix} 0 & -1 & 1 & 0.5091 \\ 0 & 0 & 1 & 0.0663 \\ -1 & 0 & 0 & 0.3218 \\ 0 & 0 & 0 & 1 \end{bmatrix} \tag{8}
$$

$$M_{s5} = \begin{bmatrix} 0 & 0 & 1 & 0.7401 \\ 0 & 1 & 0 & 0.0309 \\ -1 & 0 & 0 & 0.3189 \\ 0 & 0 & 0 & 1 \end{bmatrix} \quad M_{s6} = \begin{bmatrix} 0 & -1 & 0 & 0.9047 \\ 0 & 0 & 1 & 0.1314 \\ -1 & 0 & 0 & 0.3109 \\ 0 & 0 & 0 & 1 \end{bmatrix} \tag{9}$$

$$M_{s7} = \begin{bmatrix} 0 & 0 & 1 & 0.9860 \\ 0 & -1 & 0 & 0.1517 \\ 1 & 0 & 0 & 0.3170 \\ 0 & 0 & 0 & 1 \end{bmatrix} \tag{10}$$

However, it is important to mention that the computation effort increases per increment of the number of points calculated by PoE-FK. The SI-PoE algorithm uses these points calculated by PoE-FK to detect collisions, which could pose certain problems if few points on the robotic arm are tracked. For instance, if few points are checked for collision a small obstacle might be passing between these two points without collision. However, the physical links might be colliding with the said obstacle. Thus, the size of obstacles in the workspace is an important consideration, since both the computation effort and the collision detection accuracy of the SI-PoE directly correlate to the number of virtual points on the robotic arm that are chosen to detect collisions and avoid obstacles. If few number of virtual points are implemented, the boundary surrounding the obstacle can be increased in size to reflect collision. However, this approach constrains the joint movement even further, which would limit the search space for the SI-PoE algorithm leading to underutilized collision-free space, undesirable joint trajectories, or even an absence of the solution in critical cases. In this case, a good approach would be to choose, for instance, the CGs of all links as frames (points) for tracking, interpolate between them depending on the heuristic of relative obstacle sizes, and choose the appropriate size of boundaries surrounding obstacles. This way, the PoE-FK (Equation (3)) has to be called only for the CGs and not for virtual points between them, which considerably decreases the computational effort while maximizing the collision-free space.

The trajectory of the end-effector is generated by utilizing Bézier curves for path generation and finite jerk model for the time-scaling [29]. However, it is important to mention that the Cartesian end-effector path does not have to be a Bézier curve, and the SI-PoE algorithm can work with any Cartesian space end-effector paths. In order to satisfy the finite jerk constraint on the end-effector, a quintic polynomial time scaling of the following form is employed:

$$s(t) = 10(t/t_f)^3 - 15(t/t_f)^4 + 6(t/t_f)^5 \tag{11}$$

The plots of the time-scaling used to obtain the finite jerk trajectory are presented below in Figure 3. The finite time $t_f$ is chosen to be large enough so that the Sawyer robotic arm's joint rates are not saturated to the maximum.

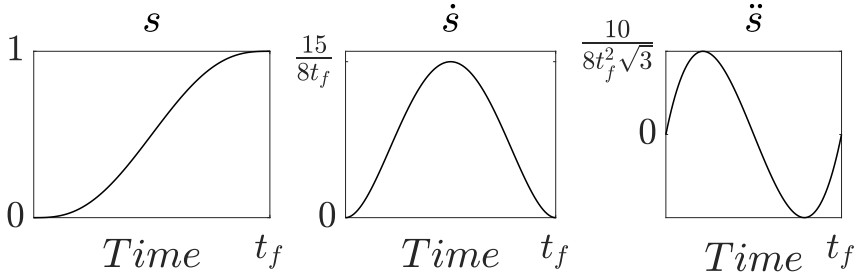

**Figure 3.** Plots of $s(t)$, $\dot{s}(t)$ and $\ddot{s}(t)$ for a fifth-order polynomial time-scaling.

An example of Bézier end-effector path is shown in Figure 4. By applying the time-scaling given in Equation (11), the trajectory now can be fully defined. The resulting trajectory is smooth and satisfies finite jerk constraint throughout the whole duration of the movement. However, it should be noted that only the end-effector movement satisfies

the finite jerk constraint, and applying a similar constraint on joint movement is shown in Section 4 (QPFJ).

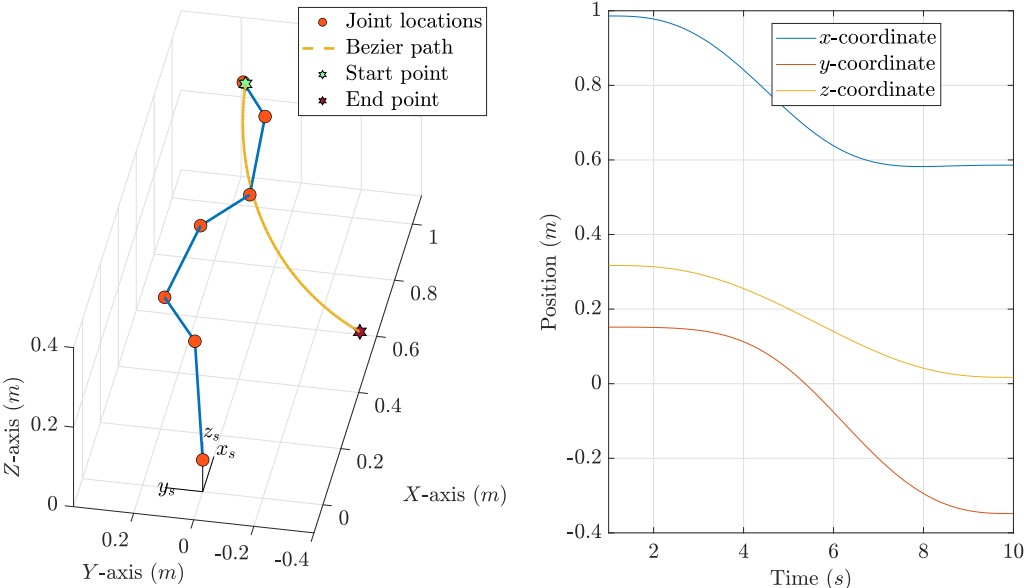

**Figure 4.** A sample trajectory generated using Bézier path and quintic polynomial time scaling.

The trajectory shown in Figure 4 was used throughout the paper as an input example to both the SI-PoE algorithm simulation and the SI-PoE Sawyer experiment. The path of the end-effector was generated using the Bézier curve, and quintic time scaling given in Equation (11) was used to produce the finite jerk profile. The SI-PoE simulation's aim is to check the algorithm's accuracy and efficiency, and the experiment's aim is to validate that the joint trajectories produced by SI-PoE can be followed accurately by the Sawyer robotic arm. The experimental torque profile obtained in Sawyer experiments was compared with the PoE inverse dynamics formulation for the Sawyer robot developed in our previous work [29].

## 3. Swarm Intelligence—Product of Exponentials (SI-PoE) Algorithm

The core of the SI-PoE algorithm is PSO and PoE. The latter was described in Section 2, while the former is a method that is a part of a larger family of SI algorithms, which describe social behavior of various ecosystems and animals such as bird flocks, schools of fish, etc. [30,31]. PSO is metaheuristic by nature, straightforward in implementation, and converges relatively quickly [32]. These are the main reasons why it is utilized in a variety of disciplines and applications. In simple terms, PSO iteratively searches through the solution space using particles. Each particle contains parameters representing a solution (fitness), which denotes its current position in the given solution/search space, as well as velocity, which influences its position (fitness), guiding it to the most optimal solution. In general, PSO is a global optimization algorithm and therefore can provide solutions within a large search space, which is very attractive when it comes to high-DoF robotic manipulators. However, PSO suffers from high computational effort when solutions to a large degree of accuracy are required. Thus, combining PSO with a relatively fast PoE-FK algorithm is proposed in this work. Furthermore, an additional advantage of combining PSO and PoE is the ease of implementation of multiple objectives due to the concise and elegant forward kinematics implementation. The SI-PoE method can be easily generalized to any $n$-DoF robotic manipulator.

The SI-PoE algorithm is initialized by assigning particles $\underline{x}_i \in \mathbb{R}^7$ to a search space (S-space) $S_{space} \in \mathbb{R}^7$. The initial location of the particles can be assigned arbitrarily or in a specific fashion which was proposed for this work.

In the proposed SI-PoE particle assignment, particles are evenly spaced in the S-space with an equal offset of $\pm 1$ rad from the "previous" solution in each (joint) coordinate, such that the swarm consists of 15 points as demonstrated in Figure 5, where each seven-dimensional swarm particle is a line. This method of particles' initialization works remarkably well due to the fact that each swarm point is essentially a change only in one of the seven joint positions.

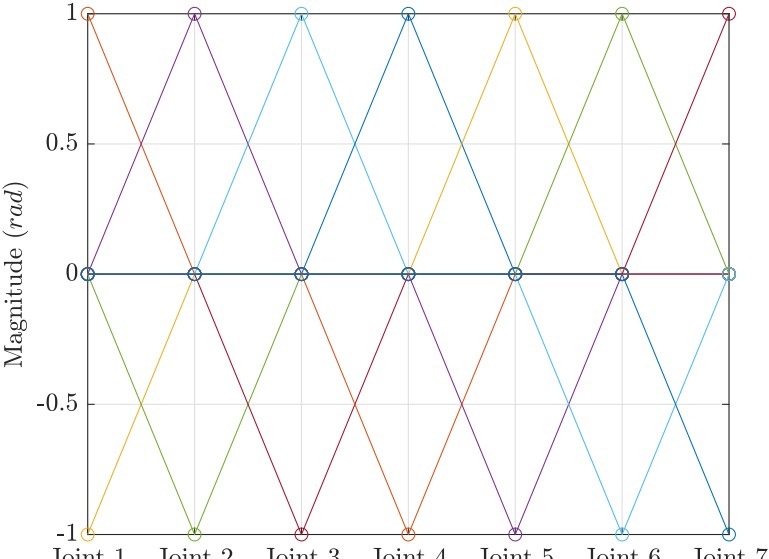

**Figure 5.** Particles $\underline{x}_i^0$ assigned to S-space starting from home configuration.

Most works choose the random method of swarm particles' initialization. However, when solving an IK problem, the known previous joint positions can serve as an initial condition or basis for the particles' distribution for each time step, which speeds up the search process while not limiting S-space. The IK computation time comparing random and the proposed particles' initial conditions is demonstrated in Table 1, where both a single end-effector position and a trajectory IK solution times are shown. The random particle's initialization was realized by adding random offsets in the range of $[-0.01, 0.01]$ rad to the previous successful joint positions. The stopping criteria were a number of iterations $(k \leq 20)$ and fitness to the desired end-effector trajectory $(f < 0.0005)$.

**Table 1.** Comparison of random and proposed particles' initialization (average of 30 runs).

| Method | 1 Point IK ($s$) | Trajectory IK ($s$) |
|--------|--------|--------|
| Random | 0.24 | 13.61 |
| Proposed | 0.21 | 12.85 |

Although computational time advantage of the proposed method may seem marginal, the real advantage becomes evident when comparing consistency of the two methods. The computation times demonstrated in Table 1 reflect the maximum number of iterations since both methods were not able to meet the fitness criterion. However, the proposed method was able to generate a trajectory with better accuracy in the same number of iterations as the random method. This is demonstrated in Figure 6, where the proposed swarm assignment has consistent small error throughout the trajectory, and the random swarm, on the other hand, exhibits large spikes of position error. It should be noted that both methods enjoy the advantage of having previously calculated joint positions, which drastically reduces the computation time and the scope of search in S-space.

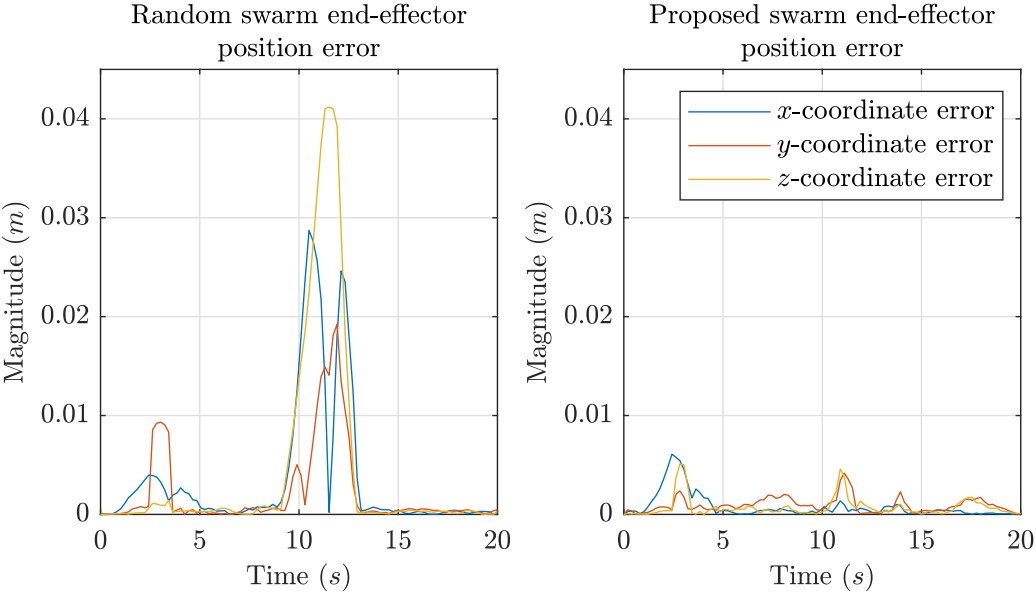

**Figure 6.** Comparison of position error of the end-effector trajectory with random swarm assignment on the left and proposed swarm assignment on the right.

As the objectives of the SI-PoE algorithm is to track pre-assigned trajectory while satisfying finite jerk, minimizing joint effort, and avoiding obstacles, the fitness function is shown in Equation (13), where both the orientation and position of the end-effector can be tracked:

$$f_i = \sigma_p \|\underline{p}_i - \underline{p}_d\| + \sigma_R \arccos\left(\frac{tr(R_i R_d^T) - 1}{2}\right) + \underline{\sigma}_J^T \mid \underline{x}_i^k - \underline{\theta}_p \mid_e + coll \qquad (12)$$

where $R_i \in SO(3)$ and $\underline{p}_i \in \mathbb{R}^3$ are computed from PoE-FK,

$$T_{si} = e^{[\mathcal{S}_1]x_i^k(1)}e^{[\mathcal{S}_2]x_i^k(2)}\cdots e^{[\mathcal{S}_n]x_i^k(7)}M_{se} = \begin{bmatrix} R_i & \underline{p}_i \\ 0 & 1 \end{bmatrix} \in SE(3) \qquad (13)$$

$$M_{se} = \begin{bmatrix} 0 & 0 & 1 & 0.9860 \\ 0 & -1 & 0 & 0.1517 \\ 1 & 0 & 0 & 0.3170 \\ 0 & 0 & 0 & 1 \end{bmatrix} \qquad (14)$$

The $\sigma_p$ and $\sigma_R$ are the weighting parameters for position and orientation, respectively, either of them can be set to zero for the cases when only the position or attitude tracking is desired; $\underline{\sigma}_J = [0.01\ 0.009\ 0.008\ 0.007\ 0.006\ 0.005\ 0.004]^T$ is the parameter penalizing excessive joint movement, where the joints closest to the base are prioritized as they move more mass; $\mid \cdot \mid_e$ is the element-wise absolute value operator; $M_{se}$ is the end-effector's home configuration; $\underline{x}_i^k$ is the $i$-th particle in the $k$-th iteration, $\underline{\theta}_p$ is the current joint positions (successful previous iteration); and *coll* is a scalar representing if any of the virtual points collide with an obstacle in the workspace if the robot assumes the $\underline{x}_i^k$ swarm particle's posture.

The collision is detected using PoE, where the number of virtual points (frames) checked by the SI-PoE depends on the obstacles' size. The virtual particles are attached to a specified link, and this would dictate their PoE-FK formula. For an arbitrary $j$-th particle that is attached to a $m$-th link, the PoE-FK can be demonstrated as:

$$T_{sj} = e^{[\mathcal{S}_1]x_i^k(1)}\cdots e^{[\mathcal{S}_m]x_i^k(m)}M_{sj} \qquad (15)$$

where $M_{sj}$ is the position and orientation of the virtual particle in home configuration given in the inertial frame.

$$coll = \begin{cases} +1, & \text{if virtual point collides with an obstacle} \\ 0, & \text{if virtual point does not collide with an obstacle} \end{cases} \quad (16)$$

After the $\underline{x}_i^0 \in \mathbb{R}^7$ particles are initialized, the velocities of the particles $\underline{v}_i^0 \in \mathbb{R}^7$ are randomly initialized with values from the $[-0.1, 0.1]$ range. The SI-PoE update law for the velocities and particles are presented in Equations (17) and (18), respectively. The numerical values of update law hyperparameters are shown in Table 2.

$$\underline{v}_i^{k+1} = \kappa[\omega \underline{v}_i^k + c_1 r_1 (PB_i^k - \underline{x}_i^k) + c_2 r_2 (GB^k - \underline{x}_i^k)] \quad (17)$$

$$\underline{x}_i^{k+1} = \underline{x}_i^k + \underline{v}_i^{k+1} \quad (18)$$

$$\kappa = \frac{2}{|2 - \phi - \sqrt{\phi^2 - 4\phi}|}, \quad \phi = c_1 + c_2 > 4 \quad (19)$$

where $\kappa$ is the constriction factor limiting the magnitude of particles' velocity, $\omega$ is the inertia weight which controls the exploration and exploitation in the search space, $c_1$ and $c_2$ are cognitive and social parameters, respectively, $r_1$ and $r_2$ are random variables with a range of $[0, 1]$, $PB$ is the best recorded individual particle's location in S-space ($\mathbb{R}^7$), and $GB$ is the best recorded global (swarm) particle location in S-space ($\mathbb{R}^7$). The SI-PoE algorithm stops if either 20 iterations were attempted or if the fitness of the current global best particle is less than the set parameter ($f_{GB} < 0.0005$). The SI-PoE algoritm steps are summarized in flowchart shown in Figure 7.

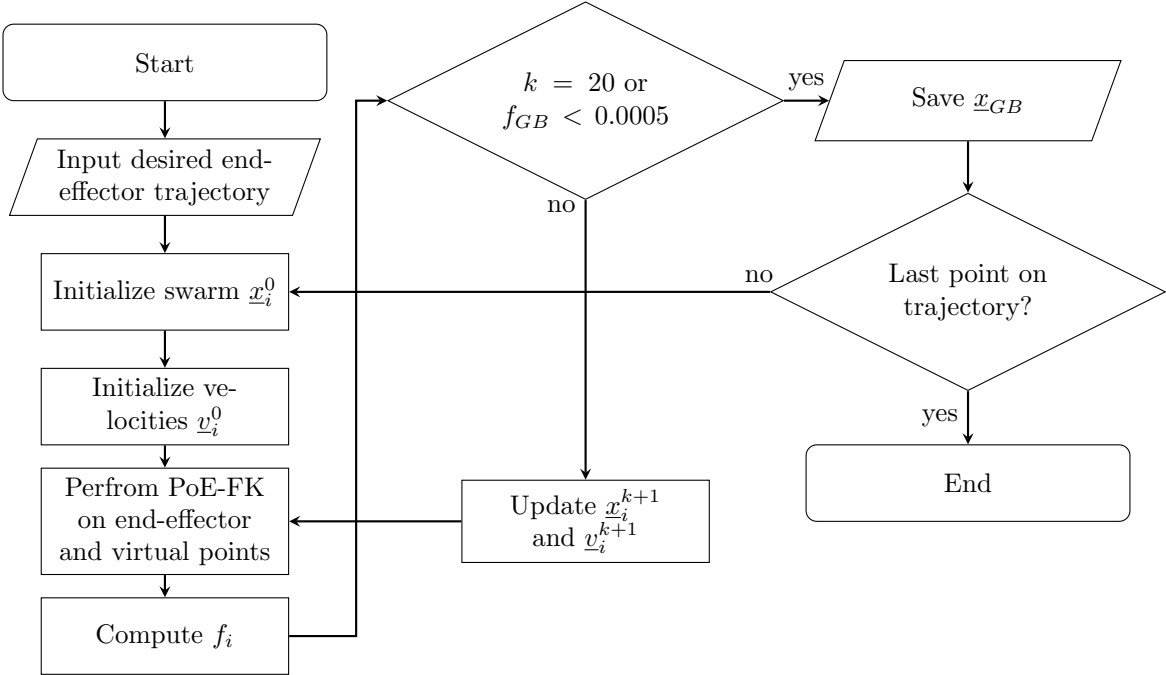

**Figure 7.** SI-PoE algorithm flowchart.

Figure 8 demonstrates an example of the swarm convergence in the S-space. The SI-PoE swarm consists of 15 points in S-space ($\mathbb{R}^7$), and it can be seen that the swarm almost converged at 50% completion. Because of the stochastic nature of the parameters within the update law, the number of iterations till convergence varied from run to run from approximately 17 to 20 iterations. However, most of the time for the cases with $\sigma_R = 0$, the SI-PoE was able to find an accurate posture of the robotic arm within 10–12 iterations with

the position error of the end-effector being less than 8–9 mm, and the rest of iterations the algorithm brought down the position error to around 5–6 mm. It was observed that the constriction parameter $\kappa$ reduces the number of iterations required to achieve said accuracy, thus positively contributing to the algorithm by shortening the computation time.

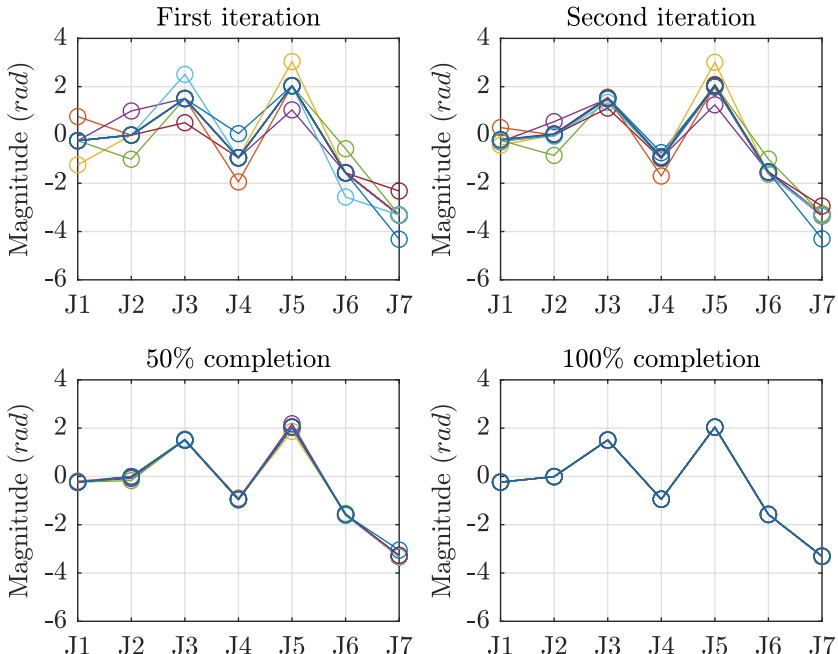

**Figure 8.** Evolution of the SI-PoE swarm.

**Table 2.** Selected SI-PoE parameters.

| Description | Variable | Value |
| --- | --- | --- |
| Initial swarm point | $\underline{x}_i^0$ | 1 rad offset at each joint |
| Initial velocity | $\underline{v}_i^0$ | random values between $[-0.1, 0.1]$ |
| Inertia weight | $\omega$ | 1 |
| Cognitive parameter | $c_1$ | 2 |
| Social parameter | $c_2$ | 2.5 |
| Random variable | $r_1$ | random value between $[0, 1]$ |
| Random variable | $r_2$ | random value between $[0, 1]$ |
| Convergence parameter | $\phi$ | 4.5 |
| Constriction factor | $\kappa$ | 0.5 |

## 4. Resulting Joint Trajectories and Quintic Polynomial Finite Jerk (QPFJ) Method for Trajectory Generation

The SI-PoE algorithm was tested on different trajectories. For the purpose of illustration and without the loss of generality, the end-effector trajectory shown in Figure 4 served as an input to the SI-PoE algorithm, and three obstacles were added to the workspace. All of the obstacles were spheres: two of them had a diameter of 20 cm centered at $[0.5, 0, 0.35]^T$ and $[0.6, -0.13, 0.4]^T$, and the third obstacle had a diameter of 40 cm with its center at $[0.5, 0, 0]^T$. The center of gravity of each link served as a virtual point for obstacle detection outlined in Equation (15), resulting in a total of 8 virtual points.

The resulting joint trajectories obtained from the SI-PoE contained short-term fluctuations, which were filtered using the moving average filter. The filtering takes negligible computational resources and takes approximately 0.02 s of computational time for a trajectory with 100 points. The raw and filtered joint trajectories along with their position error for a desired trajectory shown in Figure 4 are demonstrated in Figure 9. Applying moving average filter smooths the trajectories and eliminates spikes on the expense of slightly

increasing the position error overall that happens due to the "flattening" of the spikes. Along the trajectory, the maximum error reduces from approximately 13 mm (unfiltered) to 4 mm (filtered) at 10 s. The total error integrated over time in all 3 axes is 0.0363 ms for the unfiltered joint trajectory and 0.0440 ms for the filtered SI-PoE joint trajectory. The resulting movement of the simulated robotic arm is shown in Figure 10, where it can be seen that the SI-PoE was able to successfully generate obstacle avoiding trajectory with additional constraints such as the finite jerk on the end-effector and prioritized penalty on excessive joint movement. At approximately 10 s, the robotic arm's virtual points approach the third obstacle. In response, the robotic arm starts actively utilizing rolling joint number 5 in order to stop the movement of the robotic arm toward the obstacle and continue advancing the end-effector along the desired trajectory.

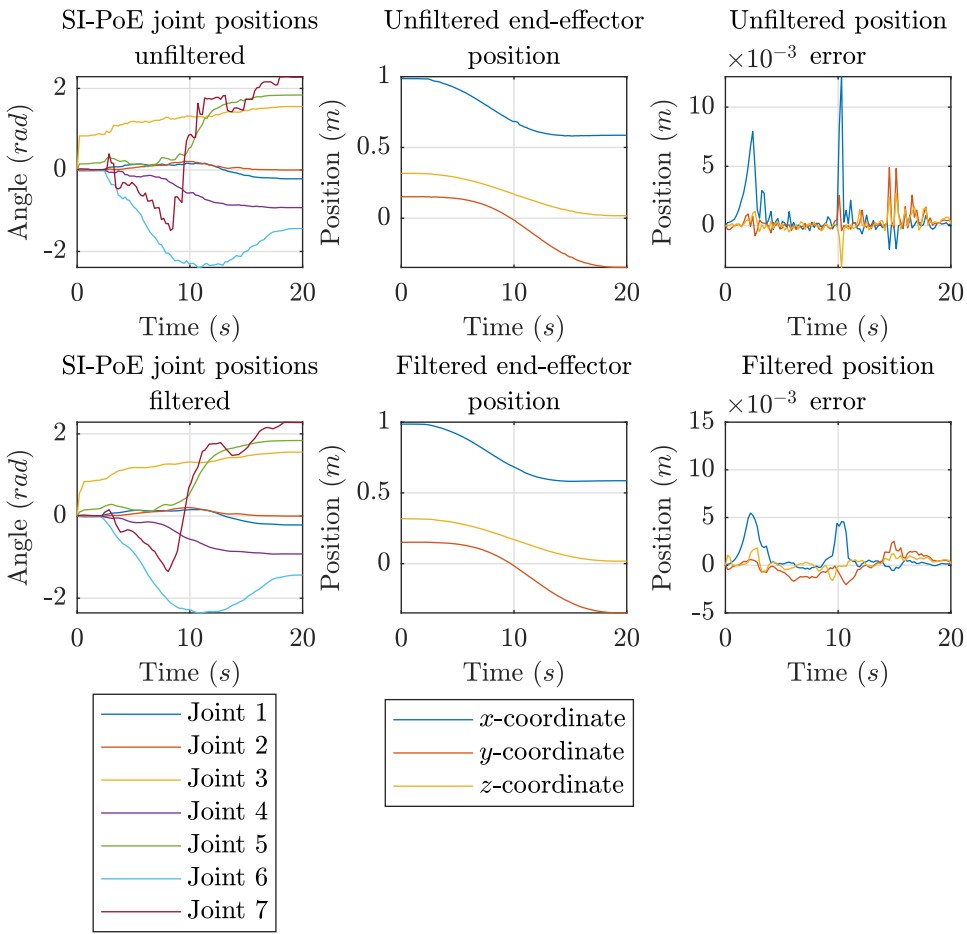

**Figure 9.** Comparison of SI-PoE unfiltered and filtered joint positions and position errors.

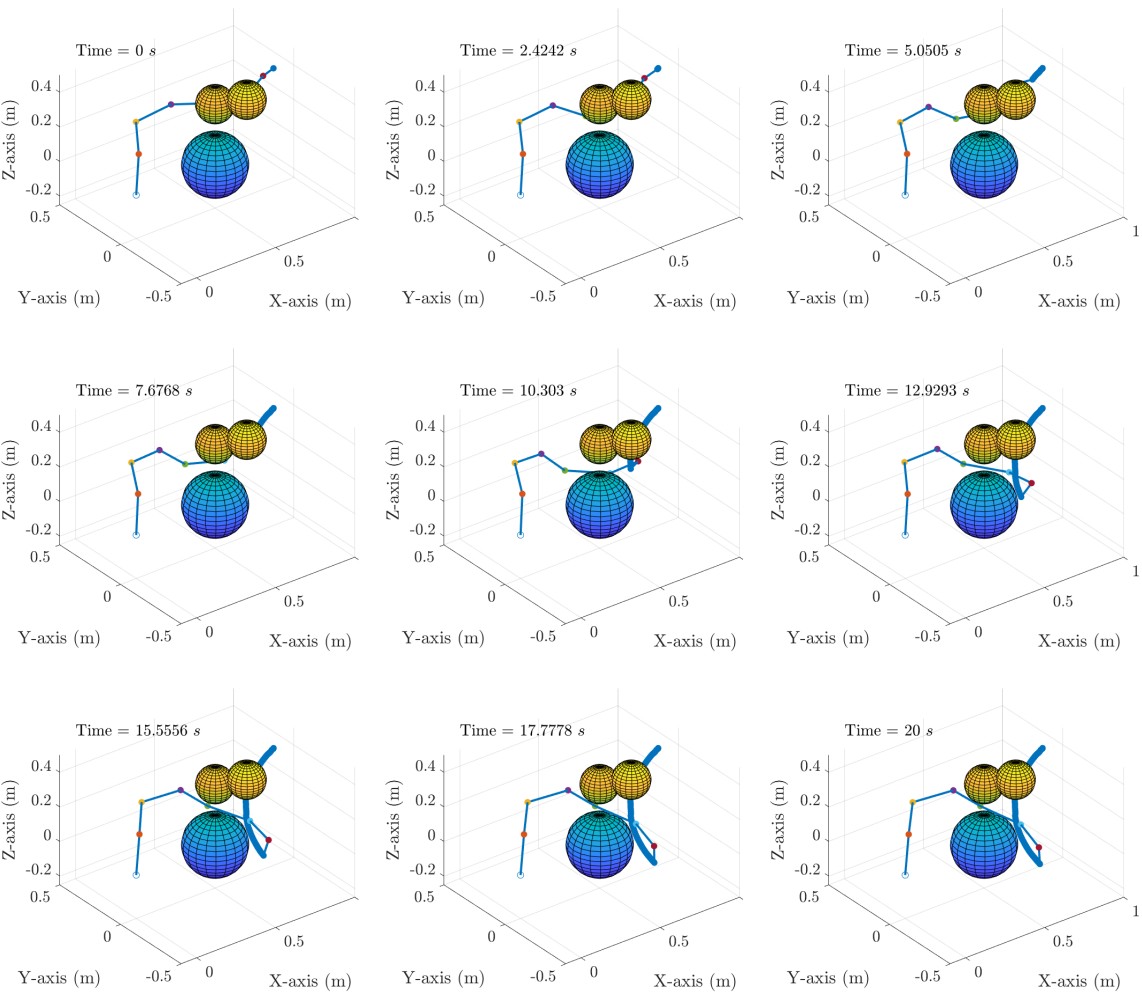

**Figure 10.** Obstacle avoidance trajectory generated using SI-PoE.

The finite jerk constraint can be applied on the joint trajectories after obtaining said trajectories from the SI-PoE or separately if the initial and final joint positions are known. Finite jerk joint positions, velocities, and accelerations as a function of time can be computed as follows using the quintic polynomial finite jerk (QPFJ) model:

$$\underline{\theta}_{\mathrm{fj}}(t) = \underline{\theta}_{\mathrm{end}}[(10/t_{\mathrm{end}}^3)t^3 - (15/t_{\mathrm{end}}^4)t^4 + (6/t_{\mathrm{end}}^5)t^5] \tag{20}$$

$$\underline{\dot{\theta}}_{\mathrm{fj}}(t) = \underline{\theta}_{\mathrm{end}}[(30/t_{\mathrm{end}}^3)t^2 - (60/t_{\mathrm{end}}^4)t^3 + (30/t_{\mathrm{end}}^5)t^4] \tag{21}$$

$$\underline{\ddot{\theta}}_{\mathrm{fj}}(t) = \underline{\theta}_{\mathrm{end}}[(60/t_{\mathrm{end}}^3)t - (180/t_{\mathrm{end}}^4)t^2 + (120/t_{\mathrm{end}}^5)t^3] \tag{22}$$

where $\underline{\theta}_{\mathrm{fj}}$, $\underline{\dot{\theta}}_{\mathrm{fj}}$, $\underline{\ddot{\theta}}_{\mathrm{fj}}$ are finite jerk joint positions, velocities, and accelerations, respectively; $\underline{\theta}_{\mathrm{end}}$ is the final joint positions obtained using SI-PoE; $t_{\mathrm{end}}$ is the total movement time.

Although Equations (20)–(22) produce joint position profile with finite jerk, the position error to the desired path becomes very high as demonstrated in Figure 11, which would certainly lead to a collision with obstacles in the workspace. One way of integrating QPFJ model to the SI-PoE is to use several via-points on the SI-PoE joint position trajectory and "sew" the trajectory from the finite-jerk profile piece-by-piece. The downside of the "sewing" approach is that it requires additional computational effort.

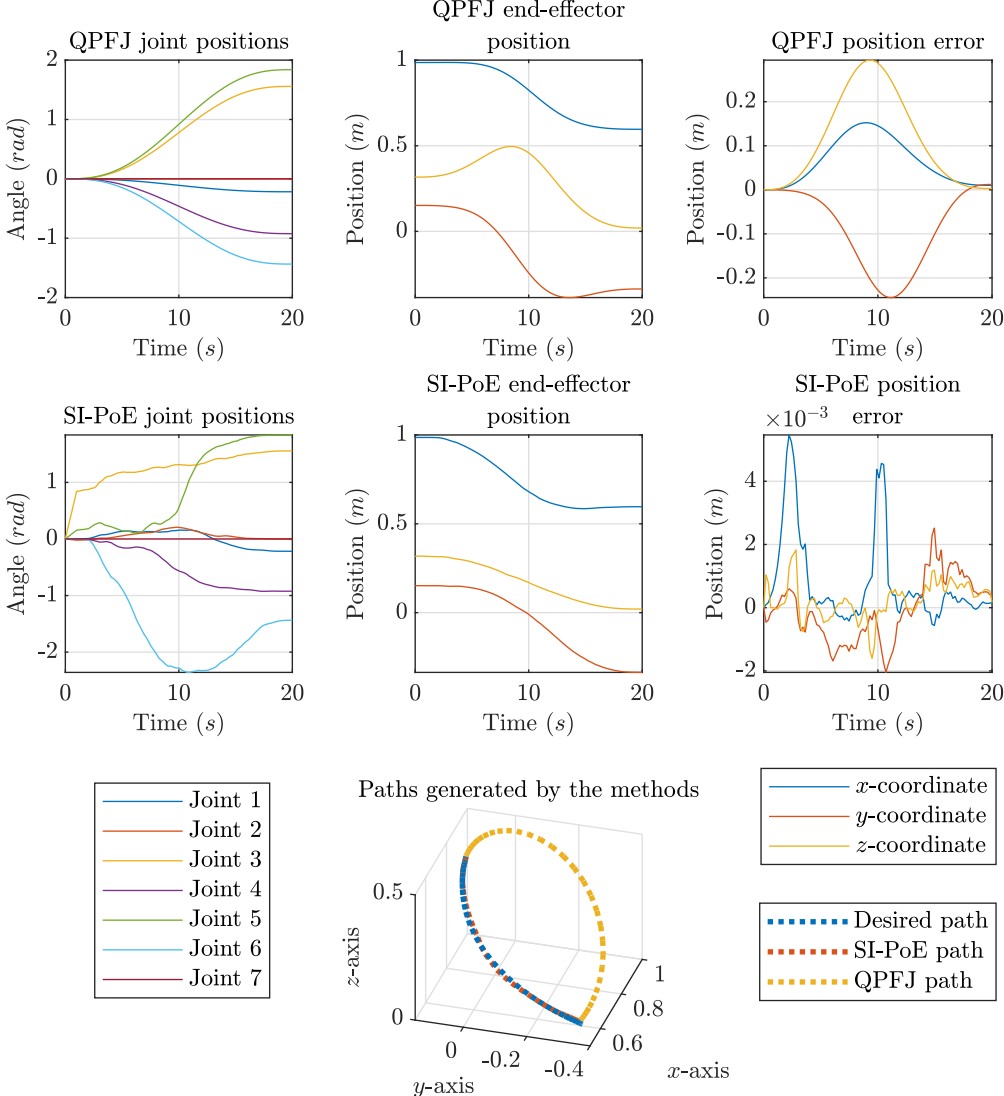

**Figure 11.** Simulated QPFJ and SI-PoE trajectories compared with the desired path.

## 5. Sawyer Experiments

The experiments were conducted on the 7-DoF Sawyer robotic arm, shown in Figure 12. For implementation and validation purposes, the joint positions obtained using SI-PoE were directly fed to the Sawyer robotic arm through a robot operating system (ROS)—Python environment on the Software Development Kit (SDK) mode. SI-PoE and QPFJ trajectores shown in Figure 11 were tested. Joint positions, velocities, and torques were recorded. The end-effector position error was calculated from the experimental joint positions.

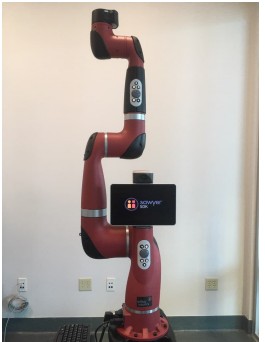

**Figure 12.** Sawyer robotic arm.

The experimental results for the joint positions and end-effector position are shown in Figure 13. The Sawyer robotic arm successfully tracked both SI-PoE and QPFJ trajectories with minimum noise introduced, which is evident in Figure 13, where the maximum position error of SI-PoE in any axis at any point of time within the trajectory is approximately 6 mm which correlates well with the simulated trajectory shown in Figure 11. The integrated error in all axes is 0.1172 ms, which is larger than the simulated integrated error of 0.0440 ms, which is attributed to the noise in the robotic arm throughout the entirety of the trajectory.

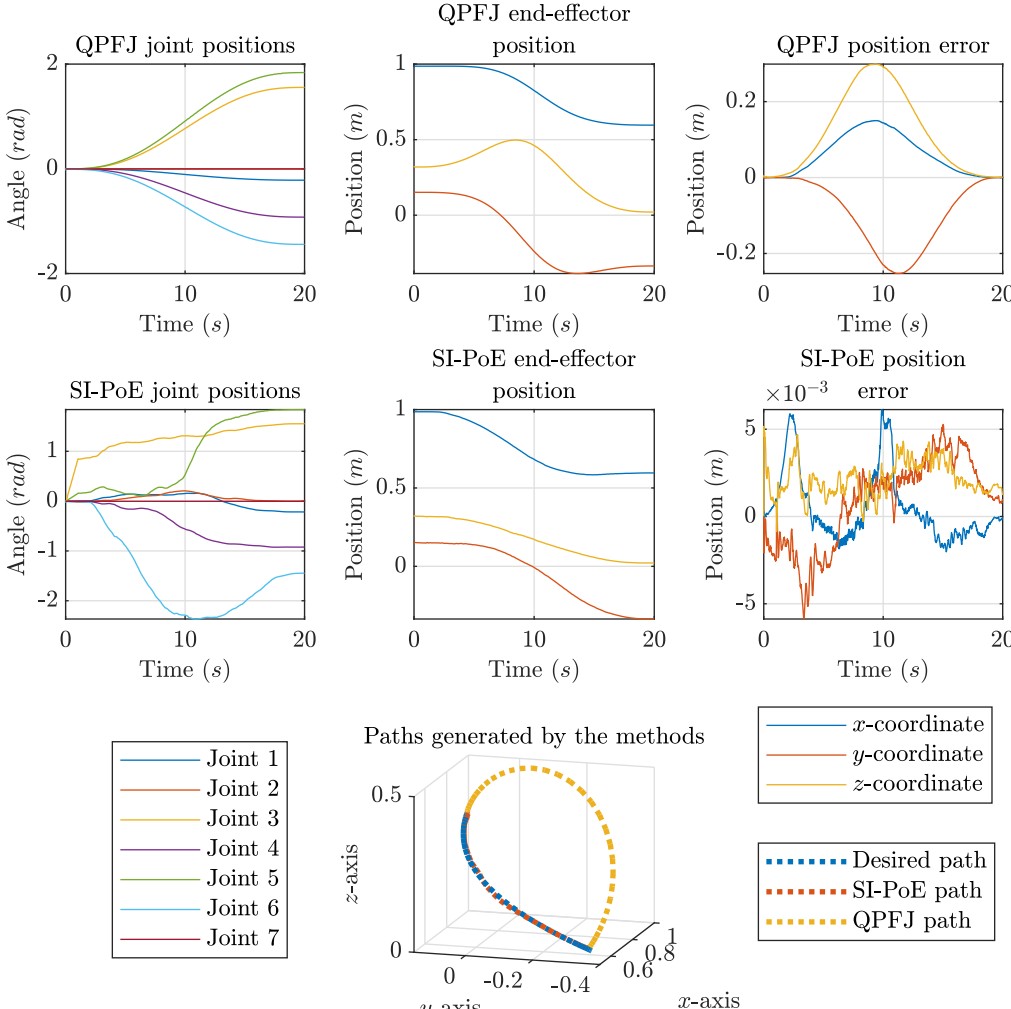

**Figure 13.** Experimental QPFJ and SI-PoE trajectories compared with the desired path.

The Sawyer end-effector tracked finite jerk profile well, while avoiding "virtual" obstacles. Eight virtual points were used for collision detection. The entire movement took 20 s, while the computation of the trajectory took approximately 13 s, indicating that the SI-PoE algorithm can be applied "online" while the robotic arm is operating, given that the number of virtual points is optimized a priori. Depending on the size of the obstacles in the workspace, the number of via-points checked for collision should be changed or the surrounding boundaries of the obstacles should be increased in size. However, such methods could lead to a limited S-space, which could result in undesired trajectories, errors in the position of the end-effector, or, in critical cases, even the absence of a solution.

The experimental torque profiles were compared with the simulated torque profiles for validation purposes. The inverse dynamics algorithm outlined in our previous work was used to generate simulated joint torque profiles [29]. Experimental torque profiles were also contaminated with noise. Nevertheless, the predicted joint torque trajectories approximate the experimental ones very well, as seen in Figure 14. It should be noted that less noise was present in QPFJ trajectories, which shows that, generally, smooth joint trajectories produce smooth torque trajectories. Torque trajectories with and without penalty on the excessive joint movement were compared. Although the proposed SI-PoE succeeded in minimizing torque by minimizing the excessive joint movement as was proposed in Equation (13), the difference was marginal with torque minimized roughly by 2–3 Nm in most of the joints.

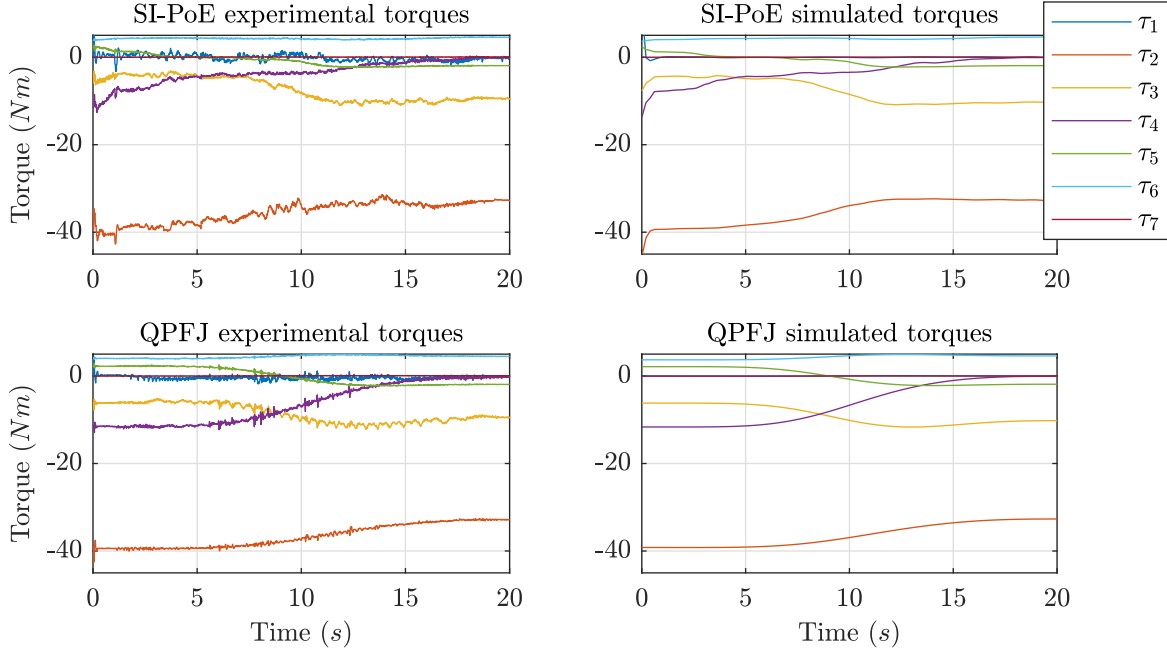

**Figure 14.** Experimental QPFJ and SI-PoE torque trajectories compared with the simulated torque trajectories.

Additionally, in order to demonstrate that diverse trajectories can be tracked by the SI-PoE algorithm, the experimental implementation was extended. First, a straight-line point-to-point trajectory satisfying the finite jerk constraint was implemented where the desired path is a straight line in Cartesian space starting at the tip of the end-effector at home configuration and ending at [0.25 m, 0 m, −0.25 m], which passes through one of the obstacles. This path was chosen on purpose to demonstrate SI-PoE's robustness. The visualization of this straight-line trajectory is demonstrated in Figure 15, where the straight-line path is shown as a green line. At the beginning, rolling joints are actuated to avoid the smaller sphere at the top at 2.5 s. In the 2.5–10 s interval, the Sawyer robot is able to move in a straight line satisfying the desired path. At approximately 10 s, the end-effector meets the collision volume. As the desired path collides with obstacle volume, the SI-PoE finds an alternative route that is close to the desired straight-line path but outside of the collision volume, which forces Sawyer to move close to the boundary of

the collision volume between 10 and 15 s. After that, the Sawyer is able to once again accurately track the desired straight-line path with minimum actuation torque.

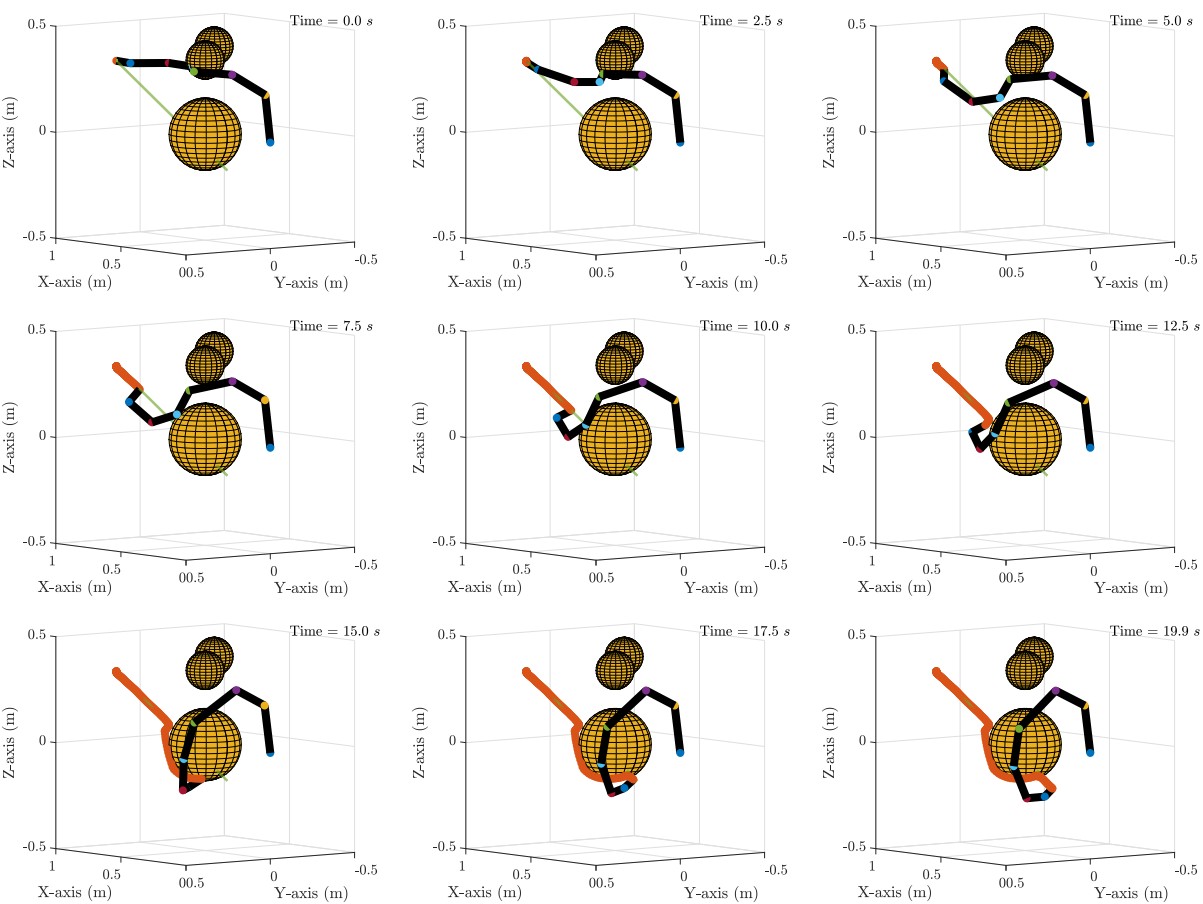

**Figure 15.** Straight-line point-to-point trajectory avoiding an obstacle generated by SI-PoE.

Furthermore, a constant screw axis (CSA) trajectory was implemented [27], with two obstacles in the way of the end-effector path. The desired path is demonstrated as a green line in Figure 16. The end-effector approaches the first obstacle at about 8 s, and in order to avoid the collision, the robotic arm moves in a clockwise direction along the collision boundary. At 10 s, the end-effector is once again on the desired path. However, at 12.5 s, the second obstacle blocks the path. This time the robotic manipulator avoids the collision by circumventing the spherical collision sphere from the top. At 15 s, the robotic arm successfully avoids the collision and continues the movement and tracking of the desired path while still avoiding collision between its links and the collision volume.

Both the straight-line point-to-point and constant screw axis trajectories were experimentally implemented on the Sawyer robotic arm. The robotic arm successfully tracked the trajectories generated by the SI-PoE algorithm. Experimental joint space trajectories, torques, and position error are demonstrated in Figure 17. As was expected, there is high error in end-effector position when the algorithm forces the end-effector to avoid collision volumes. The collision avoidance maneuver happens between 10 and 16 s in the straight-line trajectory case. For the case of the CSA trajectory, two such maneuvers are observed—the first at 8–11 s and the second at 12–16 s.

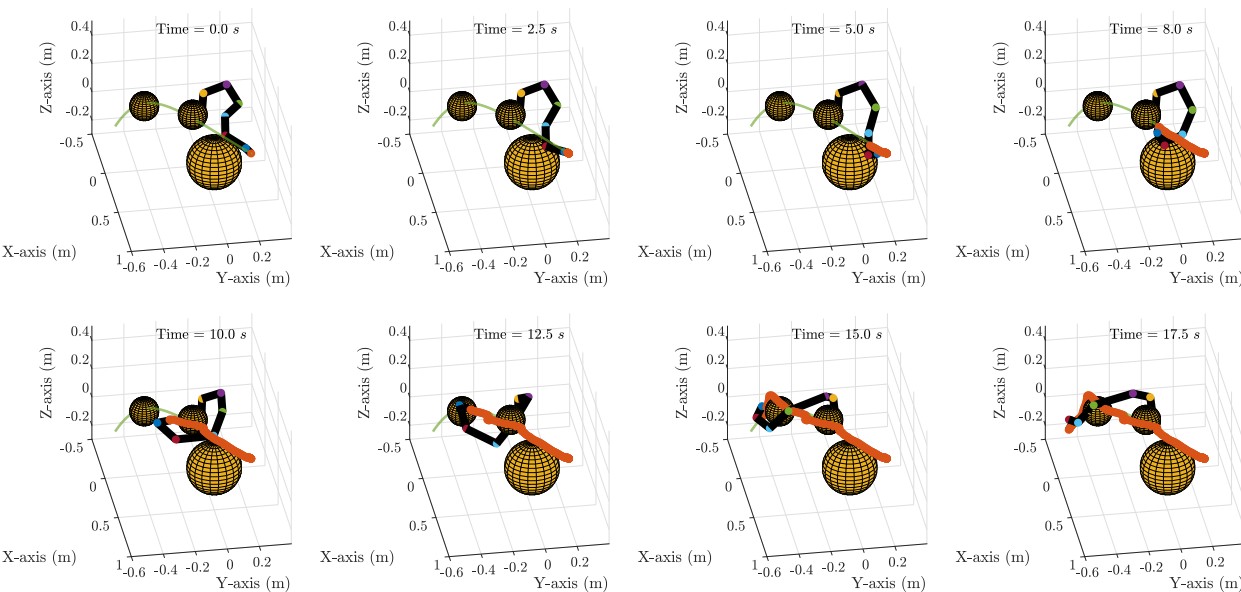

**Figure 16.** Constant screw axis trajectory avoiding two obstacles generated by SI-PoE.

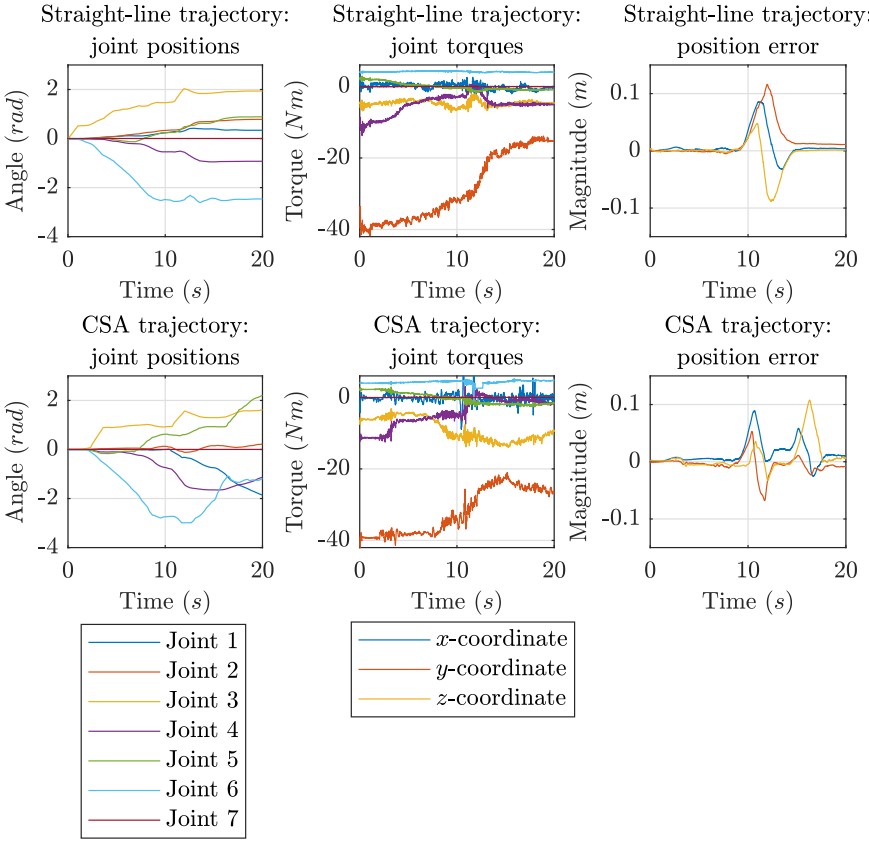

**Figure 17.** Experimental joint angle positions, joint torques, and error in end-effector position.

## 6. Conclusions

In this work, the SI-PoE algorithm was outlined, simulated, and experimentally validated using Sawyer robotic manipulator. The proposed SI-PoE algorithm utilizes PoE-FK for forward kinematics computation and collision detection. The proposed fitness function of the SI-PoE includes both orientation and position errors, penalty for excessive joint movement, and collision detection. The algorithm's hyperparameters were demonstrated

along with the proposed swarm initialization that improved both the computation time and accuracy of the end-effector position for each step. SI-PoE was able to satisfy multiple constraints such as avoiding obstacles in the workspace, minimizing excessive joint movement and subsequently minimizing torque, and tracking finite jerk end-effector trajectory as demonstrated in the simulation and experimental validation with a maximum position error of 6 mm at any time throughout the trajectory with exceptions when an obstacle is directly on the path of the end-effector. The experimental torque profiles were compared with simulated inverse dynamics torque trajectories and showed good correspondence. SI-PoE can be used "online" if the number of virtual points checked for collision is carefully picked, as was demonstrated in this work. The disadvantage of this method of obstacle avoidance is that if obstacles in the workspace are small in size, it would either require more virtual points for the SI-PoE to track using PoE-FK, which would considerably increase the computation time or increase the boundary surrounding the obstacle, increasing the effective size of the obstacles and decreasing the swarm's search space.

The finite jerk in joint space was achieved using QPFJ method, which produced smooth joint and torque trajectories. Although the finite jerk constraint was satisfied using QPFJ, the simultaneous obstacle avoidance was not achieved. The end-effector position error can be reduced by utilizing constant screw axis (CSA) trajectories. Potentially, QPFJ can be combined with SI-PoE so that the manipulator avoids obstacles and satisfies finite-jerk on its joints. However, it is expected that such a combination would increase the computational effort. An optimal combination of SI-PoE and QPFJ can be explored in future works.

**Author Contributions:** Conceptualization, A.M., T.H. and R.P.; data curation, A.M.; formal analysis, A.M., T.H. and R.P.; investigation, A.M.; methodology, A.M.; project administration, T.H. and R.P.; resources, T.H. and R.P.; software, A.M.; supervision, T.H. and R.P.; validation, A.M.; visualization, A.M.; writing—original draft, A.M.; writing—review and editing, A.M., T.H. and R.P. All authors have read and agreed to the published version of the manuscript.

**Funding:** This research received no external funding.

**Institutional Review Board Statement:** Not applicable.

**Informed Consent Statement:** Not applicable.

**Data Availability Statement:** Raw data were generated at MicaPlex, Embry-Riddle Aeronautical University. Derived data supporting the findings of this study are available from the corresponding author A.M. on request.

**Acknowledgments:** The authors would like to thank Daniel Posada for his technical support with setting up the Ubuntu-ROS-Sawyer framework.

**Conflicts of Interest:** The authors declare no conflict of interest.

## Abbreviations

The following abbreviations are used in this manuscript:

| | |
|---|---|
| DoF | Degree of Freedom |
| PoE | Product of Exponentials |
| PSO | Particle Swarm Optimization |
| IK | Inverse Kinematics |
| FK | Forward Kinematics |
| D-H | Denavit–Hartenberg |
| URDF | Universal Robot Description Format |
| CG | Center of Gravity |
| ML | Machine Learning |
| ANN(s) | Artificial Neural Network(s) |
| QPFJ | Quintic Polynomial Finite Jerk |

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
