# Peer review of "Multi-Objective Swarm Intelligence Trajectory Generation for a 7 Degree of Freedom Robotic Manipulator"

_robotics, doi:10.3390/robotics10040127_

Round 1

Reviewer 1 Report

The manuscript deploys a PSO optimization of trajectory generated using Bezier curves. Results was validated on Sawyer robotic manipulator. The solution is limited to this specific brand and cannot be applied to standard 6DoF industrial robotic arms.

The path generation and collision avoidance are well studied problems. This particle suggest method to optimize Bezier trajectory with collision avoidance. The results depend on this initial trajectory more than on its further optimization with PSO. All experiments use only single trajectory as an input for PSO optimization.

Why do you use PSO and how it compares with other optimization methods?

How do apply this method to generic 6DoF with conventional D-H parameters?

How can the algorithm compare with just simple Bezier trajectory without optimization?

Overall, I would recommend extending the experiments section with more diverse trajectories and motions. Demonstrate the benefits of PSO over other optimization algorithms.

Reviewer 2 Report

The proposed paper describes an SI-PoE method for the optimization of robot trajectories under several constraints (e.g., collision avoidance, finite jerk). The work is generally well written and easy to follow even for non-technical readers, however, there are a few fixes to be made before the publication.

In particular:

  • The literature review must be improved. For example, similar works have coped with following the trajectories [1] with very good results. Moreover, there are works that focus on trajectory generation and redundant tasks without using heuristic approaches (e.g., from Robotics [2]). Please widen that section.
  • Please explain what is e^[S1]θ1 (eq.3 and followings)
  • The "slow" computational time could be due to a slow collision detection algorithm. This method [3] could come in handy with the Sawyer robot due to the easiness of simplifying its structure. Is it the case?
  • In the figures that compare results, please use the same scale, i.e., the same axis limits (e.g., Figure 6, 7, 9). Otherwise, it is hard, at a glance, to identify which method is favorable.
  • In line 229 authors wrote "In response, the robotic arm starts actively utilizing rolling joint number 4". However, in Figure 9 no such behavior is shown. In fact, it looks like Joint 7 is being actively utilized. Do Figures 9 and 10 refer to different tests?

[1] Xidias, E.K. Time-optimal trajectory planning for hyper-redundant manipulators in 3d workspaces. Robot. Comput. Integr. Manuf. 2018, 50, 286–298.

[2] Bottin, M.; Rosati, G. Trajectory Optimization of a Redundant Serial Robot Using Cartesian via Points and Kinematic Decoupling. Robotics 20198, 101. https://doi.org/10.3390/robotics8040101

[3] Bottin M., Boschetti G., Rosati G. (2019) A Novel Collision Avoidance Method for Serial Robots. In: Gasparetto A., Ceccarelli M. (eds) Mechanism Design for Robotics. MEDER 2018. Mechanisms and Machine Science, vol 66. Springer, Cham. https://doi.org/10.1007/978-3-030-00365-4_35

Round 2

Reviewer 1 Report

Publish in present form.